# Oil Palm Breeding in the Modern Era: Challenges and Opportunities

**DOI:** 10.3390/plants11111395

**Published:** 2022-05-24

**Authors:** Jerome Jeyakumar John Martin, Rajesh Yarra, Lu Wei, Hongxing Cao

**Affiliations:** 1Coconut Research Institute, Chinese Academy of Tropical Agricultural Sciences, Wenchang 571339, China; jeromejeyakumarj@gmail.com (J.J.J.M.); rajeshyarra@rediffmail.com (R.Y.); wl79912021@163.com (L.W.); 2Hainan Key Laboratory of Tropical Oil Crops Biology, Wenchang 571339, China

**Keywords:** germplasm, particle bombardment, *Agrobacterium* tumefaciens-mediated transformation, genetic engineering

## Abstract

Oil palm, a cross-pollinated crop with long generation time, poses a lot of challenges in achieving sustainable oil palm with high yield and quality. The African oil palm (*Elaeis guineensis* Jacq.) is the most productive and versatile oil-yielding crop in the world, producing more than any other oil-yielding crop. Despite recent challenges, such as stress tolerance, superior oil quality, disease tolerance, and the need for new market niches, there is a growing need to explore and develop new varieties with high yield potential and the genetic diversity required to maintain oil palm yield stability. Breeding is an indispensable part of producing high-quality planting materials to increase oil palm yield. Biotechnological technologies have transformed conventional plant breeding approaches by introducing novel genotypes for breeding. Innovative pre-breeding and breeding approaches, such as identifying candidate genes in wild or land races using genomics tools, can pave the way for genetic improvement in oil palm. In this review, we highlighted the modern breeding tools, including genomics, marker-assisted breeding, genetic engineering, and genome editing techniques in oil palm crops, and we explored certain concerns connected to the techniques and their applications in practical breeding.

## 1. Introduction

The genus Elaeis consists of two species, namely, *E. guineensis* Jacq. and Elaeis oleifera. The center of origin and diversity of *E. guineensis* is found scattered throughout the tropical rainforests of west and central Africa, whereas E. oleifera, also known as American oil palm, is native to south and central America [1]. The most important palm oil importing countries are China, the Netherlands, and Pakistan. Palm oil makes up a significant portion of the world’s edible oils. Palm oil is extracted from the flesh or pulp of the fruit (the outer part) while palm kernel oil is extracted from the soft part of the seed (the inner part). Based on the fruit structure, oil palm is classified into dura (thick shell; less mesocarp), pisifera (shell-less; embryo rarely formed), and the commercially cultivated tenara, the D × P hybrid (thin shell; more mesocarp) with oil content [2,3]. Conventional crosses between dura and pisifera cultivars have typically improved oil palm productivity, particularly oil yield. Tenera, the F1 hybrid resulting from this cross, exhibits desirable traits from both parents [4]. Additionally, there is a high scope for increasing the oil palm yield to a greater extent. After a three-year immature period, the yield per tree increases progressively with age until it peaks around 20 years. As oil palms are typically propagated by the seed, there is a high degree of heterogeneity in the field, and the availability of sustainable production of genetic traits in traditional breeding is limited, resulting in low palm oil yield. Therefore, it is necessary to develop vegetative propagation to produce a larger number of plants from elite palms. Rapid palm growth using tissue culture is the most promising strategy to produce suitable plant materials (offshoots) and achieve high-quality, high yielding, pest-free varieties. As a result, the conception of micropropagation protocols is essential. Significant efforts have been made to improve micropropagation techniques based on either somatic embryogenesis or organogenesis [5]. Plant micropropagation is currently used for plant breeding to circumvent some of the constraints of conventional breeding, and clonal propagation of oil palm by tissue culture is widely used [6]. Plant regeneration has been achieved using callus or cell suspension cultures, and somatic embryogenesis appears to be adequate for oil palm propagation. Another technique to improve the ideal genetic traits of oil palm for the highest quantity and quality of palm oil is through genetic engineering improvements, which can aid in the introduction of the required new traits from other organisms. The application of genetic transformation paves the way for the use of a wide range of beneficial specific genes [7]. Many genetic transformation methods exist, including *Agrobacterium*-mediated transformation, microinjection, and particle bombardment, among others [8]. Each method has unique requirements that impact productivity. Although *Agrobacterium*-mediated transformation has several advantages, such as no requirements for special equipment, techniques, or protoplast culture systems, specific gene transfer to target tissues cannot be assured. For species that are not susceptible to infection by *Agrobacterium*, microinjection by delivery of specific DNA into plant cells is currently the most useful technique [9]. As a result, this strategy is anticipated to enable the transfer of desirable traits, such as quantity and quality of oil yield, to the new oil palm cultivar. However, an efficient oil palm protoplast culture system and specialized procedure are required. In general, each protoplast can form a new cell wall and later initiate cell division, followed by callus formation. This holds the potential for genetic modification of the oil palm through the use of protoplasts. Therefore, oil palm protoplasts are ideal targets for transformation by microinjection or electroporation and protoplast fusion. Recently, protoplast-based biotechnological approaches have been used to improve conventional breeding of various species, and oil palms can be improved through somatic hybridization and genetic transformation. Genome editing using artificial nucleases appears to be more efficient than conventional plant breeding methods or typical genetic engineering (transgenic or genetically engineered) methods [10]. Genome editing using artificial nucleases appears to be more efficient than conventional plant breeding methods or typical genetic engineering (transgenic or genetically engineered) methods [10]. As the potential and prospects of oil palm have been gradually recognized by all sectors of the population, including farmers, common people (consumers), and entrepreneurs, research and development on various aspects of oil palm are becoming more and more important in the country. This review provides new insights into oil palm through biotechnological tools that enable us to achieve sustainable oil palm production, as well as the ultimate success of its application in breeding improvement. 

## 2. Genetic Diversity of Oil Palm Germplasm

Germplasm collections are genetic treasures that serve as the basis for all plant-related initiatives. Despite the importance of germplasm in agricultural development programs, germplasm collection and conservation are significant in many underdeveloped nations. Crossbreeds between pisifera and dura are used to produce commercial (hybrid) tenera planting materials. For the production of tenera planting materials, the world’s oil palm seed industry has relied on a limited genetic basis. Intensive breeding and selection of Deli dura and Avros pisifera populations from Deli over several generations has resulted in the production of hybrid seeds that are genetically related and/or highly similar to several planting materials used worldwide [11]. The introduction of improved cultivars, land rivalry as a result of population growth, and the adoption of farming practices that are accelerating the destruction of genetic resources of oil palm diversity lead to the need to prevent the rapid loss of oil palm genetic resources through collection, conservation, and utilization for crop development being essential to the survival of the oil palm industry in the country. From the 1950s to the present, the primary missions of many oil palm research organizations have been to prospect, characterize, conserve, and evaluate the natural diversity of oil palm germplasm [12]. According to reports from 29 institutes, 21,103 oil palm accessions have been collected globally. Since the 1970s, the Malaysian Palm Oil Board (MPOB) has gathered 1467 oil palm accessions from wild populations, and these accessions have been conserved [13]. MPOB possesses the world’s largest collection of oil palm germplasm [14]. Therefore, oil palm germplasm accessions were investigated and discovered to contain high levels of genetic variation.

### 2.1. Oil Palm Germplasm Characterization

Oil palm germplasm characterization is the determination of the performance of materials in a collection. It refers to agronomic, morphological, and molecular fingerprints of the material in the gene bank and leads to the identification of genes encoding resistance to biotic and abiotic stress, precocity, dwarfism, productivity, and good fruit quality traits. Morphological, molecular, and biochemical approaches have been used to study genetic variation between individuals and populations. Oil palm germplasm has frequently used all three techniques, but morphological methods require extensive observation of plants in the field for real-time verification of plant performance over a long period of time [15]. 

### 2.2. Morphological Characterization of Oil Palm Germplasm

Morphological characterization will be used to examine genetic variation in oil palm by using vegetative, yield, fruit, and bunch traits. It is also used to assess the agronomic potential of oil palm germplasm and detect duplication. Commercially important characteristics, such as high oil output, superior bunch quality component, low height increment, large kernel, and long stalk, are identified. Agronomic traits, such as plant height, height increment, leaf area, leaf area index, frond dry weight, rachis length, total fronds, total leaf area, number of bunches, bunch weight, average bunch weight, fresh fruit bunch yield, single fruit weight, percent fruit to bunch, percent mesocarp, percent shell, percent kernel, are some of the parameters that are frequently evaluated under morphological techniques. The preponderance of oil palm institutions has employed these physical features to determine the genetic diversity of their oil palm collections [16]. These could be exploited to develop improved dura and pisifera fruit types that will serve as parental palm varieties for producing commercial (tenera) oil palm varieties.

### 2.3. Molecular Characterization of Oil Palm Germplasm 

Despite the fact that morphological characterization is promising, it has a number of flaws. Environmental variables may play a role in the differences, which are also time consuming and age dependent. Molecular markers, such as simple sequence repeats (SSR), random amplified polymorphic DNA (RAPD), and amplified fragment length polymorphism (AFLP), have been used to study genetic variation in oil palm [17]. Considerable genetic diversity within the MPOB germplasm collections has been reported using EST-SSR and isozyme analysis [18]. Okoye et al. [11] found high scores for heterozygosity and genetic diversity of 0.69 and 0.70, respectively, while using 10 microsatellite markers to assess genetic variety among palm parental lines. Using molecular markers to evaluate oil palm accessions addresses variance at the DNA level. DNA extraction is required prior to using the molecular marker approach. Molecular approaches are more expensive, particularly when it comes to analyzing large datasets. The study objectives, cost, and intrinsic qualities of the marker are the decisive factors in the selection of a marker approach. 

### 2.4. Characterization of Oil Palm Germplasm Using Biochemical Techniques 

Biochemicals, such as fatty acid composition, iodine value, and carotene content, are used to study genetic variability in oil palm germplasm. Noh et al. [19] studied accessions of Angolan oil palm with low lipase levels but high amounts of carotenes, fatty acid compositions, vitamin E, iodine, and oleic acid. Their findings revealed a high level of genetic variation among the accessions. The levels of biochemical markers, such as palmitic acid, were determined using a gas chromatography system and a spectrophotometer.

## 3. Role of Conventional Breeding in Oil Palm

Traditional plant breeding based on phenotypic selection has historically been used for effective crop improvement and development. Several classical breeding methods have been developed in the last century and are now used all over the world. To respond to global problems, oil palm breeders are constantly searching for new and improved palm cultivars. There are a number of tools and technologies, which, combined with our knowledge of oil palm and elite germplasm base, create two pathways for product development (Figure 1). Both pathways are deeply grounded in genetic approaches, in which we assay for a trait and then discover the underlying genes. The delineation of these genes using mapping/molecular markers identifies genes that are necessary and sufficient for a trait. Today, increasingly, both paths must come together to complete the package before a new product is developed. Conventional breeding refers to the acquisition of a hybrid new variety of oil palm from the crossing of two well-known varieties. The initial cross is followed by the selection of progeny with the wanted characteristics of resistance and for the described scope with the dwarfing phenotype. Crosses between *E. guineensis* and *E. oleifera* produce fertile hybrids, indicating the similarity between the two species. Male and female inflorescences are produced separately on the same palm, showing that the species is monoecious. This category of strategies refers to methods of crossing compatible plants, i.e., plants that would naturally interbreed without the help of the breeder. These techniques have been used since the dawn of agriculture. Two fundamental processes belong to this group: cross-pollination (breeding of a plant with another plant) and self-pollination (breeding of the plant with itself) [20]. Oil palm is a cross-pollinated plantation plant that grows gradually. Both Elaeis species are diploid, with *n* = 16 [21]. The 2C nuclear DNA content of *E. guineensis* was 3.76–4.32 pg [22], slightly lower than that of *E. oleifera* with 4.43 pg [22]. Because of their physical resemblance, oil palm chromosomes are difficult to differentiate cytogenetically. Most published research works identifying *E. guineensis* chromosomes were based on chromosomal length [23]. The whole-genome sequence of *E. guineensis* has assigned the individual *E. guineensis* chromosomes according to the size of sequence scaffolds, which correspond to the linkage group in the selected oil palm mapping population [24]. The reported length of the assembled oil palm genome was 1.535 Gb, representing some 85% of the total genome size, although only 43% of this was assembled and assigned to chromosomes. When compared to other oil crops, the *E. guineensis* genome is greater than soybean [25] and rapeseed [26] but smaller than corn and coconut [27,28]. 

In general, oil palm breeding programs are utilized to promote both dura (D) and pisifera (P) populations for the production of commercial D × P seeds. Commercial oil palm varieties are typically derived from progeny tests of crosses between selected parents. The pisifera used in hybridization are usually produced by self or crosses (tenera × tenera (T × T)), tenera × pisifera (T × P) hybridization, or pisifera × pisifera (P × P) self or crosses. On the other hand, the dura is selected from the dura × dura (D × D) autologous or cross. In this scenario, the selected dura palm trees offspring can continue to the next cycle [17]. Breeders are focused on developing cultivars with high fresh fruit (FFB) yields and higher mesocarp content to increase palm oil productivity. An optimal GS method should provide the highest possible accuracy, limit overfitting of the training set, and be based on marker QTL LD rather than relatedness. In addition, such approaches must be easy to implement, consistent across a variety of functions and datasets, and computationally efficient. Perennial plants can benefit more from GS than annual plants [29]. Oil palm breeding as we know it uses the selection process to create new varieties of plants with the intention of improving the genetic potential of oil palm species. Commercial variations are developed around the world by pollinating Deli dura mother palms with African pisifera pollen, and a recurrent selection strategy appears to be particularly appropriate for oil palms. Conventional breeding approaches have been highly effective in modifying the FA profiles of edible oil crops to develop commercial oils due to the unique and exotic nature of many industrially relevant fatty acids (FA). Oil palm cultivation takes about 12 years for the first generation and 40 years for the next eight generations. This is accomplished by using a vast planting area due to oil palm’s open pollination behavior. As a result, the development of novel oil palm traits or variants via conventional breeding is significantly delayed. Unlike the common practice with annual plants, it is impossible to produce homozygous parental families because of the length of oil palm reproduction cycle and strong inbreeding depression. Several traits, such as dwarf palms and plants high in vitamin E and oleic acid, have been developed. On the other hand, the oil palm has a serious problem with incomplete dominance inheritance. For example, the tenera fruits displayed all three fruit forms, dura:tenera:pisifera, in a 1:2:1 ratio, indicating the incomplete dominance of dura over pisifera. That is, only 50% of the fruits retained tenera properties. This problem makes oil palm propagation by seed germination unsatisfactory.

## 4. Importance of Tissue Culture in Oil Palm 

Oil palm cannot be propagated by vegetative tissue culture, as it has a single growing apex with no base branches. Oil palm as a monocot grows from a primary shoot meristem and has no apparent morphological basis for vegetative reproduction. Therefore, oil palm tissue culture is unique, undergoing callus and embryogenesis processes and rigorously tested in the 1960s and 1970s. Until recently, no substantial attempt was made to develop tissue culture programs for the propagation of elite palm. Some attempts have been made with seedling or embryo conditions described in regeneration protocols [6]. Therefore, oil palm tissue culture is entirely dependent on somatic embryogenesis [30]. As a result, in vitro regeneration of mature elite palms or tissue culture is the default method to proliferate an elite palm, and hence, the most essential component in oil palm development. The procedure of oil palm tissue culture is separated into many stages: (i) Sampling of explant from selected ortet; (ii) Callus initiation in explants; (iii) Embryogenesis; (iv) Shoot and root regeneration; (v) Transplanting of plants from ex vitro condition. However, the long process of solid culture medium (52–55 months) resulted in a significant proportion of oil palm abnormalities [31], which were mostly attributed to somaclonal variation. In order to improve the frequency of somaclonal variation, embryogenic structures are commonly disseminated to enhance SE efficiency [32]. Under certain circumstances, embryogenic cells and non-embryogenic cells can only be distinguished by morphological traits and a number of potential genes with differential expression between embryogenic callus. These include a putative Aux/IAA gene and Eg707, an unidentified protein implicated in abscisic acid production [33]. Although highly efficient transformation systems have been achieved in oil palm species, most transgenic oil palms are still generated using transformation protocols based on tissue culture methods, which are genotype independent, time consuming, and laborious (Figure 2). Major improvements have been made in many tissue cultures, notably in oil palm suspension culture, where the process has been shortened by at least 35 months. Therefore, oil palm suspension culture has become a valuable starting material for oil palm genetic engineering to accelerate the transformation process [34]. 

## 5. Genetic Engineering of Oil Palm

Numerous techniques for transferring genes into plant cells have been developed, and ongoing attempts are made to improve efficiency. Genetic engineering is perhaps the most effective way to address the limitations of conventional breeding, which would be difficult or impossible to achieve with traditional methods. The ultimate focus of these efforts is the development of a transgenic oil palm with a high oleic acid content [35]. MPOB pioneered the use of genetic engineering to grow palm trees with a high oleate content for the industrial commodity and liquid oil industries. When oleic acid content is >65%, the estimated value of high oleate palms is USD 1500/ha/year [36]. Oil quality modification, such as stearic acid increase, value-added oil, such as palmitoleic acid and ricinoleic acid synthesis, and novel products, such as biodegradable polymers, have been studied. The development of a reliable transformation and regeneration mechanism is of crucial importance for genetic engineering. Currently, biolistic and *Agrobacterium*-mediated transformations are commonly used to introduce useful genes into oil palm (Table 1). The parameters influencing the biolistics for oil palm transformation, such as the appropriate selectable marker [37], promoter selection [38], and ideal physical and biological factors [39], have been determined. This bombarded several hundred embryogenic calli with numerous genes, including those involved in the fatty acid biosynthetic pathway, to enhance oleic and stearic acid [40] and polyhydroxyvalerate (PHBV) genes to produce biodegradable polymers [41].

*Agrobacterium*-mediated transformations have also been used to improve the oil palm through genetic engineering. However, oil palm is not a natural host for *Agrobacterium*-like dicots. Nevertheless, several studies have been conducted to improve this method, including the use of immature embryos (IE) [42] and immature zygotic embryos (IZE) [43] as target tissues, as well as the parameters that affect *Agrobacterium*-mediated transformation of embryogenic oil palm calli [41]. The Bacillus thuringiensis insecticide protein gene (Bt) [44], the cowpea trypsin inhibitor gene (CpTI) [45], which responds to insect pest, and the PHB gene [46] have all been transmitted by *Agrobacterium*-mediated oil palm transformation. The successful regeneration of transgenic oil palm relied on selectable markers. Genes encoding hpt (hygromycin phosphotransferase), pat or bar (Phosphinothricin acetyltransferase), nptll (neomycinphosphotransferase), and EPSPs were the most often adopted selective markers for crop transformation [47]. These marker genes are implemented efficiently for selection of *Agrobacterium*-transformed cells when controlled by constitutive promoters, such as the 35S promoter from cauliflower mosic virus and ubiquitin promoters [48]. Using the *A. tumefaciens* strain LBA4404, the plasmid pBIDOG with the DOGR1 gene under the control of the 35S promoter was converted into oil palm embryogenic calli [49]. 

A longer selection process for plants developed from callus formation and somatic embryogenesis tends to result in chimeric transgenic plants more frequently. It has been suggested that both methods still require optimization of DNA delivery and selection of transformants to overcome problems associated with escapes. To overcome problems related to biolistics and *Agrobacterium*-mediated transformation, an alternative transformation method is urgently needed, and transformation of oil palm protoplasts using DNA microinjection is a promising approach. Masani et al. [39] developed unique transformation methods based on polyethylene glycol-mediated transfection and DNA microinjection to show that protoplasts are good targets for genetic manipulation of oil palm. Intense electrical pulses cause transient porosity to develop in the cell membrane, allowing molecules, including DNA, RNA, proteins, antibiotics, and pigments, to be transported into the cell [50]. Although protoplasts were the most common target for electroporation in plant research, multiple investigations showed that the technology could also be utilized to transfer genes directly into plant tissues. Some researchers have reported that utilizing linear plasmid DNA rather than circularized plasmid DNA, as well as infusing spermidine into the incubation buffers, has an effect on transformation efficiency. The incorporation of new traits into established species can reduce the time required for genetic improvement in oil palm. The appearance of a blue spot in the plant tissue as a result of GUS activity toward the X-Gluc substrate may indicate that the transformation technique was effective. The gene has also been used to express transitory genes in various types of oil palm explants, including somatic embryos [51]. Similarly, Chris Darmawan [52] reported that embryogenic oil palm calli were transferred using a vector, plasmid pCAMBIA 1303 driven by the 35S promoter carrying the GUS gene, and that transformation was successful. This system relies heavily on effective plant transformation and regeneration techniques. The main advantage of the approach is that the tissues are kept in the same physiological state after electroporation as before the transformation.

**Table 1 plants-11-01395-t001:** Genetic transformation of oil palm through *Agrobacterium*-mediated transformation and direct DNA transfer methods.

Explants	Transformation/ Strain	Vector	Selection Marker	Promoter Used	Reporter Gene	Transgene/ Expression	Studied Parameters	References
Young Leaves	LBA4404	*pUBA*	bar	Ubi1	Gus	Southern Bolt	Study glufosinate-ammonium- resistant transgenic oil palm	[41]
Embryonic Calli	LBA4404	pBIDOG	*DOG*R*1*,	CaMV35S	Gus	PCR/ Southern Bolt	Introduction of new selection agent	[49]
Immature Embryo	LBA4404	pCAMBIA 1310	nptII	CaMV35S	Gus	Gus Assay	In vitro culture of IE for direct plant regeneration	[42]
Protoplast, Basidiospore, mycelium	LBA4404	pCAMBIA 1300	hyg	gpd fungal promoter	Gus/gfp	PCR	Understanding pathogenicity factor associated with *G.boninense*	[53]
Calli Clump	Electroporation	pCAMBIA 1303	hptII	CaMV35S	GusA/ mgfp5	PCR	An efficient electroporation-mediated transformation method for oil palm calli	[52]
Embryonic Calli	Biolistic	pMI11, pMI11G, pMI3, pMI3G	pmi gene	Ubi1, CaMV35S	GusA	PCR/ Southern Bolt	To produce transgenic oil palm by using mannose as a selectable agent	[38]
Immature Embryo	Biolistic	pBI121	npt II	CaMV35S	Gus	Gus Assay	In vitro culture of IE for indirect plant regeneration	[42]
Embryonic Calli	Biolistic	pBINPLUS	hpt	35S (2XCaMV35S)	GFP	PCR/ Southern Bolt	Increase the level of transgene expression and transformation efficiencies in oil palm	[54]
Immature Embryo	Biolistic	pPSP’AP-VF6	HYG	CaMV35S, MSP	-	PCR	Successful integration of antiPATE driven by MSP in American oil palm plantlets	[43]

## 6. Association Mapping for QTL Identification

In the MAS approach, molecular markers are used for indirect selection of challenging qualities, such as traits that are not phenotypically visible at the seedling stage [55]. Molecular markers are DNA strings that are distributed across the plant genome and may be tracked as they segregate through generations. Because known genes of interest are linked together, the presence of the marker ensures the presence of the gene of interest and allows the breeder to track its segregation throughout breeding. Because breeders can recognize the markers in seedlings and do not need to acquire adult plants for selection, MAS speeds up the process of conventional plant breeding. Furthermore, MAS facilitates the enhancement of traits that cannot be easily determined using conventional methods [55]. Depending on the markers, these procedures range from gel electrophoresis, PCR, through contemporary high-throughput genotyping techniques. MAS has become a standard phase in the breeding of most crops and provides several benefits, including the increasing knowledge about molecular markers, genes of interest, and their localization. Biological traits are often governed by either a large number of genes with minimal effects (polygenic inheritance), termed as quantitative traits, or by single genes with significant effects (monogenic inheritance), referred to as qualitative traits. Quantitative traits in plants have been targets of crop improvement for yield, quality, and pest resistance. Quantitative traits are associated with QTL loci, containing genes on a chromosome. Genes that regulate quantitative traits interact with one another, and the expression of these traits is often influenced by environmental conditions. To study polygenic characteristics, quantitative trait locus analysis is utilized. Multiple potential parameters, including fruit weight, petiole cross-section, axial length, and shell:fruit, mesocarp:fruit, and kernel:fruit ratios have been empirically proved. MPOB also executed QTL research on oil palm oil quality, discovering 11 QTLs in four distinct linkage groups with iodine value C14:0, C16:0, C16:1, C18:0, C18:1, and C18:2, using a frame diagram comprising AFLP, RFLP, and SSR markers [56]. Using an interspecies pseudo-backcross of *E. guineensis* and E. oleifera revealed 19 QTLs pertaining to palm oil fatty acid content [2]. MPOB is Malaysia’s leading oil palm research and development center. MPOB has carried out extensive genetic diversity studies that can quantify the genetic distance between different breeding materials, assisting in the identification of new elite material suited for introgression into breeding programs. Using progenies derived from a self-pollinated *E. guineensis* palm (Palm T128), they constructed a linkage map and identified an RFLP marker for Sh gene [56]. Four distinct SNP markers were found to be mapped on either side of the Sh gene, with the closest marker (SNPM00310) at a distance of 2.2 cM. Gan et al. [57] used high-density SNP and DArT markers, with SSR markers as anchor loci, to construct genetic linkage maps of two closely related oil palm populations, which enable the discovery of markers strongly associated with the ‘shell thickness’ (Sh) gene. When the population is biased by the natural population and other important criteria (such as high-resolution QTL mapping, more alleles, less time, and a larger reference population), association mapping provides numerous advantages. However, association mapping in oil palm is uncommon. Using 200K SNP to perform GWAS on the oil-to-dry mesocarp content of 2045 genotyped oil palms, 80 loci were determined to be substantially correlated with oil-to-dry mesocarp yield (P104), and three important signals were determined [58]. They described the most comprehensive use of high-density SNP genotyping and GWAS methodologies to identify SNP variations correlated with differences in important yield parameters in oil-to-dry mesocarp and further confirmed their involvement in independent crosses.

## 7. Functional Markers and Their Application

Molecular markers have been found to be an effective technique for assessing genetic diversity and elucidating genetic relationships among species. Molecular markers, such as RAPD, AFLP, and SSR, have been utilized to simulate genetic diversity coupling and association mapping, as well as the construction of linkage maps. Two important monogenic hereditary traits of oil palm are the gene for fruit color (Vir) and the gene for shell thickness (Sh). Two RFLP markers associated with the fruit color gene were identified in the MPOB linkage map: MET16 (3 cM) and KT3 (4 cM) [59]. It was discovered that the markers are able to distinguish not only the Nigrescens and Virescens fruits but also the homozygous and heterozygous variants of the Vir fruit. Due to the lack of carotenoids in the exocarp, Vir fruits are green when immature and bright orange when ripe. Many molecular approaches have been employed to study the shell thickness gene (Sh) and/or marker(s). Mayes et al. [60] developed an oil palm RFLP linkage genetic map using a population derived from a self-pollinated tenera palm, which segregated by shell thickness, allowing the discovery and mapping of an RFLP marker (pOPgSP1282), which is distantly connected to Sh by 9.8 cM. This marker is quite far from Sh to be used to identify crop species, as the probability of recombination between the marker allele and the gene is still high. RAPD markers have the potential to be utilized to genotype both parents and hybrids [61]. Singh et al. [62] identified two mutations in the MADS box transcription factor of the SHELL gene, which is homologous to the Arabidopsis thaliana gene, SEEDSTICK (STK), which controls ovule identity through homozygosity mapping. They also detected an SNP in the 28 or 30 codon that prevents shell (Sh) from binding to DNA normally, resulting in a shell-less phenotype. RAPD markers have been used to evaluate the genetic diversity of natural palm oil species, such as E.oleifera species [63]. In another study, RAPD work by Moretzsohn et al. [64] on linkage mapping of the shell thickness locus also showed two different markers, R11-1282 and T19-1046, which were 17.5 cM and 23.9 cM, respectively, on either side of the sh+ locus. Although the two markers were even further removed from the Sh gene, the authors claimed that using a flanking marker-based assay would allow tenera and pisifera palms resulting from a D × P cross with an error rate of only 4% (0.175 × 0.239 = 0.042). Therefore, with these two markers, a more accurate and faster identification of the fruit shape is possible. Dou et al. [65] recently performed genetic variation analyses on a panel of twenty oil palm accessions to find single-nucleotide polymorphisms (SNPs) associated with vitamin E content. The BLASTP analysis revealed 37 similar genes in oil palm corresponding to the vitamin E biosynthetic pathway, with lengths ranging from 426 to 25,717 bp (average 7089 bp). Within the Arecaceae family, the utility of transcribed libraries and cDNA tag archives has resulted in the development of EST-SSR in oil palm [66]. Other investigations on Cocos nucifera and P. dactylifera found similar findings [67]. Although most of these methods are unfeasible due to the high cost of implementation, they have become obsolete as the utility of simple sequence repeats/microsatellites has evolved. Billotte et al. [68] discovered a high-density microsatellite-based backtracking map of oil palms. It is the first linkage map with 16 independent linkage groups corresponding to the 16 homologous chromosome pairs of oil palm. Using 944 SSR and AFLP markers, this integrated map spanned 1743 cM of the plant genome. It was discovered that E-AGG/M-CAA132, an AFLP marker, maps at 4.7 cM from the Sh locus and is positioned near the terminus of LG4. Recently, Babu et al. [69] discovered a cleaved amplified polymorphic site (CAPS) marker for oil palm fruit type differentiation. The CAPS identifier will enable the selection and timely delivery of attractive high-yielding variants. For oil palm, Bai et al. [70] identified two potential QTL that are related to foliar surface, and this trait is strongly related to oil yields. This would allow breeders to select high-oil-yielding trees indirectly through MAS by identifying trees with a large leaf area and reduce the laborious and destructive phenotyping required to measure oil efficiency.

### Genomic Selection of Oil Palm

For quantitative traits, the efficiency of QTL-based marker-assisted selection (MAS) is limited because it overestimates the effect of strong QTLs and fails to exploit weak QTLs [71]. To overcome these issues, genomic selection was developed. Oil palm is a diploid, monoecious, allogamous plant with a high GS potential. Reciprocal recurrent selection (RRS) is the most used method for improving oil palm genetics [72]. The RRS is based on two populations: the Deli and Group B (African origin). In each of the two populations, full-sib families were sampled, and progenies were examined in Deli × Group B crosses. Genomic selection has the potential to rapidly evolve heterotic pools in plants where they are not well established, such as oil palm. When selecting single crosses, genomic prediction could reduce the need for test crosses and field evaluations [73]. RRGS programs have been initiated in oil palm, which has a long generation interval and high phenotyping costs [74]. The genomic feature modeling approaches were designed to improve the prediction of complex traits, provided the groups of selected markers were enriched for causal variants. The potential of genomic models to forecast the genetic value of unevaluated selection candidates was exhibited using SNPs and without optimizing the training and validation populations, with prediction accuracies ranging from 0.14 to 0.73 for various yield components [74]. The GS model predicted the performance of unevaluated hybrid crossings better than a control model utilizing pedigree data instead of markers for five yield components (FFB, O/M, BN, BW, and M/F) [74]. This emphasized GS’s ability to capture genetic variations within full-sib families, as well as genetic differences across families, allowing for the selection of the best individuals from the best families, as is now performed with phenotypic breeding schemes.

## 8. Genome Editing Technology

The utilization of the CRISPR/Cas9 system in genome editing has found wide application in various fields, such as medicine and agriculture [75]. The versatility of this system has been applied from knockout or edits of a gene, regulating gene expression until single nucleotide polymorphisms arise [76]. In plant genome editing, two currently used delivery methods of the CRISPR/Cas9 system are particle bombardment and *Agrobacterium* transformation. The CRISPR/Cas9 module is simpler than ZFNs and TALENs and is regarded as a breakthrough technology for genome editing. The module is based on Cas9 nuclease and a modified RNA single guide (sgRNA) for editing target nucleotide sequences [75]. This could be efficiently exploited in the oil palm to modify desirable genes by targeting specific DNA sequences with tailored nucleases containing sequence-specific DNA-binding domains. Recently, a novel technique based on the CRISPR-Cas9 system, termed ‘base editing’, has been developed. Most recently, Yarra et al. [77] proposed the notion of using base editing techniques in oil palm. It is HDR independent and edits single DNA bases at precise places in the genome without causing DSBs or adding a foreign DNA template. Although base editing has been effective in improving rice and wheat yields, its application to other crops is still in its early stages. In oil palm, the CRISPR/Cas9 transgenes can be delivered via young leaves, and stably transformed plants can be regenerated via tissue culture. Bohari et al. used rice as a model system in order to develop a genome editing system for oil palm. The genomic sequences of rice and oil palm were discovered to have considerable similarities [78]. This might be utilized efficiently in oil palm to modify desirable genes by targeting specific DNA sequences with tailored nucleases containing sequence-specific DNA-binding domains. Transcription activator-like effector nucleases were utilized to downregulate two fatty acid desaturase 2 genes (FAD2-1A and FAD2-1B) in soybean, resulting in an increase in oleic acid content due to a decrease in linoleic acid content [79]. Even though it is a great approach with a multitude of benefits, it could be used in oil palm to more effectively regulate specific genes or to replace an undesirable gene. With the advent of oil palm genome sequencing, this has now become feasible.

## 9. Conclusions

Unlike other oil crops, palm oil also has the most sustainability certification schemes. In 2004, a non-profit organization called the Roundtable on Sustainable Palm Oil (RSPO) was formed. The RSPO has become a leading certifier, developing a set of environmental and social criteria for the production of certified sustainable palm oil (CSPO). In other words, the level of rigor around the sustainability of palm oil is unprecedented. Because of this, the expansion of cultivation through deforestation is now banned in many countries, and fertile land is being reduced by urban development. Therefore, many research organizations had initiated the efforts to improve oil yield on the existing land bank to meet increasing demand. On the other hand, selective breeding for high-yielding, stress-tolerant materials is also equally essential in the long run; however, the breeding cycle of perennial palm oil is about 12 years per generation. Therefore, improving oil palm yield through conventional breeding is time consuming, while increasing production through simply expanding acreage is unsustainable. The application of the diagnostic and analytical power in modern biotechnology would be a great advantage in oil palm breeding programs. There is a heavy commitment in terms of labor for traits maintenance and recording. All these factors result in high costs of breeding programs. It is clear that biotechnology offers unprecedented opportunities to improve plant breeding. Biotechnological application offers researchers new insights and tools that improve their efficiency and effectiveness. This stabilizes the broad phenotypic variability of the heterozygous hybrids, resulting in recognized palm cultivars with consistent traits. The use of molecular tools and transgenic crops will allow us to meet our food needs in a sustainable manner with limited availability of resources. Therefore, the development of biotechnological tools to maximize the progression of genomic resources in oil palm is essential for the continuous improvement in oil yield and the overall quality of oil produced, as well as the production of market-responsive varieties, which are essential for long-term agricultural sustainability.

## Figures and Tables

**Figure 1 plants-11-01395-f001:**
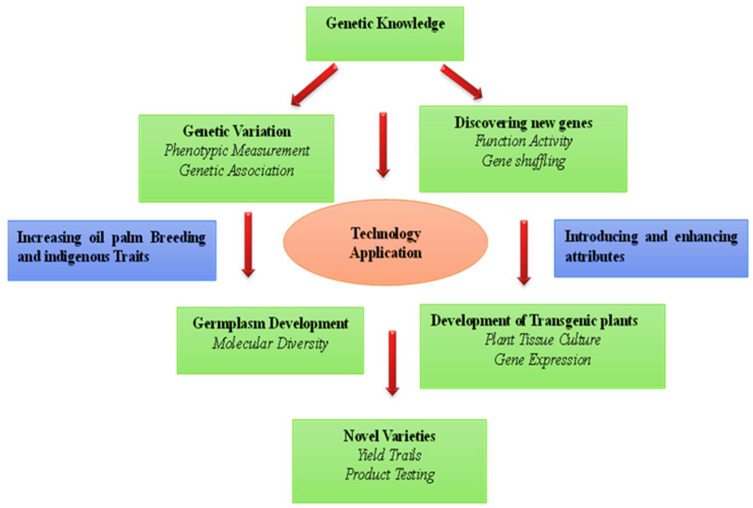
Tools for oil palm breeding. The left side exhibits the identification of native traits and molecular isolation of genes that reside in the oil palm, although sometimes in wild variants or in low-yielding varieties. The right side depicts discovery of transgenic traits, where we want to change the expression levels, location, or timing of a gene, or to use a gene from a different species.

**Figure 2 plants-11-01395-f002:**
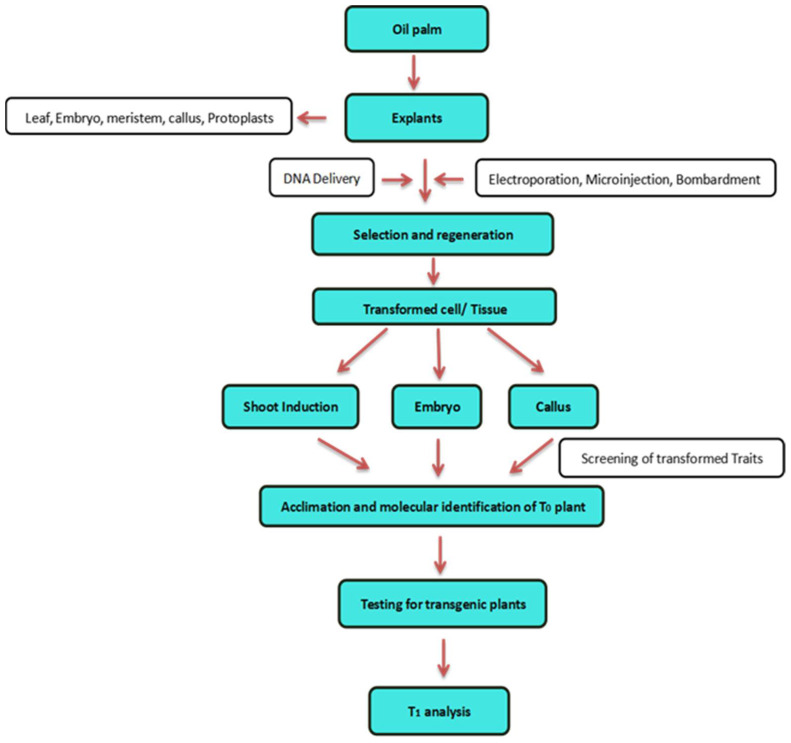
A summary of the technique for producing transgenic oil palm utilizing the direct DNA delivery method.

## Data Availability

Not applicable.

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
