# Peer review of "Oil Palm Breeding in the Modern Era: Challenges and Opportunities"

_plants, 2022, doi:10.3390/plants11111395_

Round 1

Reviewer 1 Report

a) The review is simple but effective. b) It would be nice if the authors could elaborate more on the following statement: "However, oil palm has a serious issue with 121
inadequate dominance inheritance." in lines 121-122. c) Nothing is mentioned about the number of chromosomes and ploidy level of the Elais species.  I think it would be important to include that information.

Author Response

Dear Reviewer,

Reviewer 2 Report

The authors summarized and reviewed the breeding strategies of conventional and biotechnological methods in oil palm breeding and their applications. It has important significance for researchers in related fields. I suggest that authors would better to supplement or introduce some existing reviews and new literatures, and make some comparison with the existing reviews, in order to highlight the novelty and value of this paper, such as "Mat Yunus Abdul Masani, Abang Masli Dayang Izawati, Omar Abdul Rasid, Ghulam Kadir Ahmad Parveez, Biotechnology of oil palm: Current status of oil palm genetic transformation, Biocatalysis and Agricultural Biotechnology, 2018, Vol.15, pp 335-347","Maizura Ithnin, Wendy T. Vu, Min-Gyoung Shin, etc., Genomic diversity and genome-wide association analysis related to yield and fatty acid composition of wild American oil palm, Plant Science, 2021, Vol. 304, 110731" and so on, which are the latest progress.

Author Response

Dear Reviewer,

Reviewer 3 Report

General comments

The paper "The role and contribution of oil palm in breeding and biotechnology - challenges and opportunities" presents an interesting but incomplete review on breeding and biotechnology on palm oil.

Oil palm breeding has been carried out in different countries with different selection methods such as recurrent selection, see for instance:

Baudouin, L., Baril, C., Clément-Demange, A., Leroy, T., & Paulin, D. (1997). Recurrent selection of tropical tree crops. Euphytica, 96(1), 101-114.

Gomes Junior, R. A., Freitas, A. F. D., Cunha, R. N. V. D., Pina, A. J. D. A., Campos, H. O. B., & Lopes, R. (2021). Selection gains for the palm oil production from progenies of American oil palm with oil palm. Pesquisa Agropecuária Brasileira, 56.

Good genetic gains were obtained, and new selection methods are now developed, like genomic selection, see:

Cros, D., Denis, M., Sánchez, L., Cochard, B., Flori, A., Durand-Gasselin, T., ... & Bouvet, J. M. (2015). Genomic selection prediction accuracy in a perennial crop: case study of oil palm (Elaeis guineensis Jacq.). Theoretical and applied genetics, 128(3), 397-410.

Wong, C. K., & Bernardo, R. (2008). Genomewide selection in oil palm: increasing selection gain per unit time and cost with small populations. Theoretical and Applied Genetics, 116(6), 815-824.

Tissue culture is also an important avenue in the oil palm breeding strategy, but the methods face the problems of somaclonal variants, see:

Jaligot E., Wei Yeng Hooi, Debladis E., Richaud F., Beulé T., Collin M., Agbessi M.D.T., Sabot F., Garsmeur O., D'Hont A., Syed Alwee S.S.R., Rival A. 2014. DNA methylation and expression of the EgDEF1 gene and neighboring retrotransposons in mantled somaclonal variants of oil palm. PloS One, 9 (2) : 14 p.

Rival A., Ilbert P., Labeyrie A., Torres E., Doulbeau S., Personne A., Dussert S., Beulé T., Durand-Gasselin T., Tregear J., Jaligot E. 2013. Variations in genomic DNA methylation during the long-term in vitro proliferation of oil palm embryogenic suspension cultures. Plant Cell Reports, 32 (3) : p. 359-368.

Specific comments

The title is not correct in my opinion; the subject is rather “The role of breeding methods and biotechnology on oil palm cultivation” (or on oil palm performance)

Abstract

The abstract is a little too long and very general. It could be interesting to address the specific challenges facing oil palm breeding.

Introduction

Line 29: “The centre of origin and diversity of E. guineensis seems to be in the tropical rainforests of west and central African.", not “seems to be”, but “is”; it is well-known; se for instance:

Maley, J. (2001). Elaeis guineensis Jacq.(oil palm) fluctuations in central Africa during the late Holocene: climate or human driving forces for this pioneering species? Vegetation history and Archaeobotany, 10(2), 117-120.

Line 37: Pisifira (with capital), change in all the text

Line 43:  I do not agree with the text“…. resulting in low palm oil yield.”, and it is not in accordance with the abstract you wrote above “….is the most productive and versatile oil yielding crop in the world, producing more than any other oil yielding crop.” (lines 10-11).

Lines 45- 48: There are not only advantages to developing clonal crops, especially for disease resistance; see for instance:

Gibson, A. K. (2021). Genetic diversity and disease: The past, present, and future of an old idea. Evolution.

The mastery of tissue culture is necessary to develop research on genetic transformation, and tissue culture is still confronted with the problems of somaclonal variants.

Role of conventional breeding in oil palm

Lines 91-92: I do not understand the sentence “When crossed with E. guineensis and oleifera, it produces a homologous pairing, indicating that the two species are identical.”. The two species are not identical!

Fig. 1 is not really understandable.

Importance of tissue culture in Oil palm

Lines 147-149: “Therefore, oil palm suspension culture has become a valuable starting material for oil palm genetic engineering to accelerate the transformation process”, where is the reference, or the results?

Genetic engineering of Oil palm

This part is like a catalog, but it is not clear what was applied on oil palm.

The part “Genetic diversity of oil palm Germplasm” should be placed at the beginning, after the introduction.

Functional markers and their application

In this part, it would be interesting to consider the genomic selection.

Genome editing technology

You have to cite the paper:

Yarra, R., Cao, H., Jin, L., Mengdi, Y., & Zhou, L. (2020). CRISPR/Cas mediated base editing: a practical approach for genome editing in oil palm. 3 Biotech, 10(7), 1-7.

The conclusion is very general and does not really take into account the specificities of oil palm.

Author Response

Dear Reviewer,

Reviewer 4 Report

The manuscript summarizes some interesting research on oil palm breeding and the available breeding technologies.

I have the following comments.

The title does not represent the manuscript text. It does not convey any scientific notion and it  is very general. I suggest that the authors state clearly the objectives or goals of this review and revise the title accordingly.

I believe that the most serious drawback of this review is that it is very theoretical and does not provide practical applications. The manuscript does not discuss or present data on the different technologies used for oil palm breeding. Authors, you will need to present data on the form of figures or tables and justify your conclusions based on data and practical applications.

For example, stating that biotechnological tools will enable us to achieve sustainable oil palm production is vague since the authors do not present supporting evidence. Is the oil palm production higher when you use biotechnological versus conventional breeding tools? If yes, how much higher? Is the cost higher? Authors need to address these concerns and more.

I suggest that the authors revise and rewrite this review by eliminating vague statements and focus on the data from practical applications and technologies of oil palm breeding.

Author Response

Dear Reviewer,

Round 2

Reviewer 3 Report

General comments

The paper "An overview of the importance of breeding and biotechnology for sustainable improvement in oil palm" has been improved in comparison to the first version. However, there are still several points that need to be improved, and the English has to be checked by a native English.

Specific comments

Lines 42-43: the end of the sentence: “….and the availability of sustainable genetic trait production in traditional breeding is limited.” is not really clear; a reformulation is necessary.

The part “Role of conventional breeding in oil palm” needs modifications. You cite general articles like:

van de Wiel, C.; Schaart, J.; Niks, R.; Visser, R. Traditional plant breeding methods.Wageningen, Netherland: Wageningen UR 541 Plant Breeding 2010.

but you do not cite specific articles about palm selection programs like:

Baudouin, L., Baril, C., Clément-Demange, A., Leroy, T., & Paulin, D. (1997). Recurrent selection of tropical tree crops. Euphytica, 96(1), 101-114.

Or

Yusop, M. R., Sukaimi, J., Amiruddin, M. D., Jalloh, M., Swaray, S., Yusuff, O., & Chukwu, S. C. (2020). Genetic improvement of oil palm through recurrent selection. In The oil palm genome (pp. 35-46). Springer, Cham.

It is indeed important to specify that a great number of varieties currently distributed are issued from reciprocal recurrent selection programs; and it would also have been interesting to have an idea of the genetic progress that has been achieved by classical selection, see for instance:

Nyouma, A., Bell, J. M., Jacob, F., & Cros, D. (2019). From mass selection to genomic selection: one century of breeding for quantitative yield components of oil palm (Elaeis guineensis Jacq.). Tree Genetics & Genomes, 15(5), 1-16.

Lines 160-161: The sentence “When crossed with E. guineensis and oleifera, it produces a homologous pairing, indicating similarity between two species” is not clear. I suppose you wanted to indicate “Crosses between E. guineensis and E. oleifera produce fertile hybrids indicating the similarity between the two species.”, and I am wondering if similarity is the god word.

Line 209: “…which is nonetheless typical with annual plants.”? E guineensis is not an annual plant.

Line 211 : “Nevertheless, dwarf palms and plants with high 211 vitamin E and oleic acid content have been developed.”, what is the link with the previous sentence?

And “However, oil palm has a serious issue with inadequate dominance inheritance.”: badly written.

All the part needs an important improvement.

Fig. 1 is not really understandable; it is important to improve it in order to clarify the message

Line 222 : why “unique”

Lines 226-227: I suppose you want to mean: “Oil palm cannot be propagated vegetatively, except by tissue culture because it has a single growing apex with no base branches.”

Please read carefully in order to correct the text; there are too many approximate sentences.

In the conclusion, it is necessary to be more careful and work towards other perspectives. Indeed, the only message that genetic transformation can make oil palm cultivation more sustainable is not very convincing and can be criticized; see for instance “RSPO - Roundtable on Sustainable Palm Oil”, https://rspo.org/, and:

Cattau, M. E., Marlier, M. E., & DeFries, R. (2016). Effectiveness of Roundtable on Sustainable Palm Oil (RSPO) for reducing fires on oil palm concessions in Indonesia from 2012 to 2015. Environmental

Reviewer 4 Report

The authors have addressed some of my comments but not all of them.

First, English language and syntax needs major improvement throughout the text.

Authors, you will need to present data on the form of figures or tables and justify your conclusions based on data and practical applications.

Is the oil palm production higher when you use biotechnological versus conventional breeding tools?

If yes, how much higher? Is the cost higher?

Please address these concerns and more.

Round 3

Reviewer 3 Report

General comments

The paper "An overview of the importance of breeding and biotechnology for sustainable improvement in oil palm" has been improved and now acceptable.

Author Response

Dear Reviewer,

Thank you for your acceptance of our manuscript entitled “An overview of the importance of breeding and biotechnology for sustainable improvement in oil palm”. Those comments are all valuable and very helpful for revising and improving our paper, as well as the important guiding significance to our researches. Once again, thank you very much for your acceptance.

Thank You.

Reviewer 4 Report

Authors, this is still a very theoretical review...

In the author's response, you should let the reviewer know where you made the changes by listing the page no. and line numbers. You letter states that "we have made the changes accordingly". But this is very general and forces the reviewer to spend time trying to figure out where the changes are.

I suggest revising the title to better reflect the text. I suggest the following title:

Oil palm breeding in the modern era: challenges and opportunities 
